# Identification of Novel Susceptibility Genes for Early-Onset Colorectal Cancer Through Germline Rare Variant Burden Testing

**DOI:** 10.3390/cancers17243931

**Published:** 2025-12-09

**Authors:** Ruocen Song, Reger R. Mikaeel, Zhongping He, Mehgan Horsnell, Wendy Uylaki, Weimin Meng, Nicola K. Poplawski, Bernd Wollnik, Yun Li, Jinghua Feng, Hamish S. Scott, Yufeng Shen, Chen Wang, Rui Yin, Yousong Ding, Xavier Llor, Wendy K. Chung, Eric Smith, Timothy J. Price, Joanne P. Young, Xiao Fan

**Affiliations:** 1Department of Biomedical Engineering, University of Florida, Gainesville, FL 32611, USA; 2Department of Pathology and Immunology, Washington University School of Medicine, St. Louis, MO 63110, USA; 3Department of Biostatistics, University of Florida, Gainesville, FL 32611, USA; 4Department of Haematology and Oncology, The Queen Elizabeth Hospital, Woodville, SA 5011, Australia; 5Department of Health Outcomes and Biomedical Informatics, University of Florida, Gainesville, FL 32611, USA; 6Adult Genetics Unit, Royal Adelaide Hospital, Adelaide, SA 5000, Australia; 7Adelaide Medical School, University of Adelaide, Adelaide, SA 5000, Australia; 8Institute of Human Genetics, University Medical Center Göttingen, 37075 Göttingen, Germany; 9Centre for Cancer Biology, SA Pathology and University of South Australia, Adelaide, SA 5001, Australiahamish.scott@sa.gov.au (H.S.S.); 10Department of Genetics and Molecular Pathology, SA Pathology, Adelaide, SA 5000, Australia; 11Faculty of Health and Medical Sciences, University of Adelaide, Adelaide, SA 5000, Australia; 12Department of Systems Biology, Columbia University Irving Medical Center, New York, NY 10032, USA; 13Department of Biostatistics, Columbia University Irving Medical Center, New York, NY 10032, USA; 14Department of Medicinal Chemistry, Center for Natural Products, Drug Discovery and Development, University of Florida, Gainesville, FL 32611, USA; 15Internal Medicine, Yale School of Medicine, Yale University, New Haven, CT 06520, USA; 16Department of Pediatrics, Boston Children’s Hospital, Boston, MA 02115, USA; 17Harvard Medical School, Boston, MA 02115, USA; 18University of Florida Health Cancer Center, Gainesville, FL 32610, USA

**Keywords:** colorectal cancer, early-onset colorectal cancer, germline variants, rare variant burden test, oncogenes, tumor suppressor genes

## Abstract

The genetic mechanism for colorectal cancer is complex, involving multiple risk genes, each contributing to a small proportion of cases. Our study sought to clarify this complexity, with a focus on early-onset colorectal cancer. We identified several novel colorectal cancer-susceptible genes, advancing our understanding of the disease etiology. These findings lay the groundwork for future functional validation studies and may ultimately inform the expansion of clinical gene panels for colorectal cancer.

## 1. Introduction

Colorectal cancer (CRC) is the third most frequently diagnosed malignancy and the second leading cause of cancer-related mortality worldwide, with an estimated 1.9 million new cases and nearly 904,000 deaths reported in 2022 [1]. While CRC is traditionally associated with aging populations, the incidence of early-onset colorectal cancer (EOCRC) has been rising in recent decades across multiple countries. Although the exact definition of EOCRC varied with age cutoffs of 30, 35, 40, 50, or 55 years in different studies, the incidence of this malignancy has been rising across all of these age groups [2,3]. In adults < 55 years old, the incidence of CRC rose by ~20% from 1994 to 2014 [2]. EOCRC exhibits distinct biological and clinical characteristics, often presenting at more advanced stages and with unique molecular features compared to CRC in older individuals. Despite these alarming trends, the risk factors contributing to EOCRC remain incompletely understood [3].

CRC is traditionally classified as sporadic or hereditary, with nearly 30% of EOCRC patients having a first-degree relative with CRC. Current reports suggest that germline pathogenic/likely pathogenic (P/LP) variants occur in 9% to 26% of EOCRC cases, depending on study design, inclusion criteria, and genetic testing methodologies, with 10% carrying P/LP in genes associated with colorectal and polyposis syndromes and 2.5% harboring P/LP variants in other cancer-related genes [4]. The genetic underpinnings of the majority of EOCRC are still unclear. Identifying genetic risk factors is crucial for mitigating cancer risk, guiding targeted prevention, optimizing therapy selection, assessing individuals at risk, and advancing research on novel therapeutic and preventive strategies.

Advances in sequencing technologies enable comprehensive exploration of rare germline coding variants and their role in disease susceptibility. By aggregating rare deleterious variants into gene-based burden tests, statistical power is enhanced, facilitating the detection of genetic contributors to cancer risk [5]. Here, we conducted a case–control association study using rare variants from exome sequencing data to identify novel EOCRC risk genes. Given the observed heritability and role of rare variants in cancer susceptibility, we hypothesized that EOCRC risk genes should have an enriched burden of deleterious variants in EOCRC cases.

## 2. Materials and Methods

### 2.1. The South Australian Young Onset Colorectal Polyp and Cancer Study (SAYO) Case–Cohort

The SAYO investigated risk factors and warning signs for CRC/significant polyps in individuals under 55 years old [6]. The study, approved under ethics HREC/14/TQEHLMH/194, has enrolled 270 participants from 2015 to 2022 through interviews and medical record reviews. SAYO participants underwent risk assessments, including evaluations of family history, colorectal polyps, and type 2 diabetes mellitus. Among the 270 participants, 193 were diagnosed with CRC, and 44 had significant polyps only [7]. Given that significant polyps are recognized precursor lesions to CRC and may share overlapping germline susceptibility factors [8,9], the inclusion of patients with polyps provides critical insights into early disease diagnosis and increases the power to detect rare variant associations. Blood samples were collected for exome sequencing [10]. Mismatch repair (MMR) deficiency status was determined as previously described [11].

### 2.2. The Simons Foundation Powering Autism Research for Knowledge (SPARK) Control Cohort

We used the parents of probands with autism in SPARK [12] as controls for our analysis. There was no information about cancer history in these parents. We utilized the SPARK-integrated exome sequencing v2 dataset.

### 2.3. Individual Quality Control—Ancestry and Relatedness Predictions

The SPARK and SAYO relatedness predictions were completed using Kinship-based INference for Gwas (KING) [13]. If second-degree or closer relatives were detected within the cohort, we retained only one representative per family. Ancestry estimation was conducted using Peddy [14], which projects samples based on principal component analysis (PCA) derived from the 1000 Genomes reference populations and assigns ancestry labels based on population clustering. We visualized the first two principal components colored by the predicted racial and ethnic groups among our SAYO cases. As 94.4% of our cohort are genetically inferred to be of European ancestry, we restricted our analysis to individuals of European ancestry. The same criteria were applied to the SPARK dataset.

### 2.4. Variant Calling and Annotation

SAYO patients underwent exome sequencing using the KAPA HyperPrep Kit, the Roche SeqCap EZ MedExome Enrichment Kit, and the Illumina NextSeq 500 (2 × 150 bp paired-end reads [15,16]. Sequencing reads were aligned to the hg38 reference genome using Burrows-Wheeler Aligner (BWA) version 0.7.17 [17], and variant calling was performed using the Genome Analysis Toolkit (GATK) version 4.1.9.0 [18,19]. For both the SAYO and SPARK datasets, we used Variant Effect Predictor (VEP) version 113.0 [20] to annotate variant types based on Ensembl [21,22,23] and population frequencies based on gnomAD v3.1 [24] and UK Biobank (UKBB) (accessed August 2025) [25]. We included missense and predicted loss-of-function (LoF) variants for risk gene analysis [26]. LoF variants included premature stop/start-gain, stop/start-loss, frameshift, and splicing variants. To assess the deleteriousness of missense variants, we applied the Variant Effect Scoring Tool version 4 (VEST4) [27], which quantifies the functional impact of variants based on evolutionary conservation and functional parameters.

### 2.5. Variant Quality Control

We applied quality control (QC) metrics to eliminate potential false variant calls and mitigate technical biases between the SAYO and SPARK datasets, ensuring the validity of our analytical approach. As detailed in Appendix A, we required the following QC thresholds: a minimum genotype quality of 10, allele depth for alternative alleles of 2, a genotype depth of 7, and allele balance of 0.1. Variant Quality Score Log-Odds (VQSLOD) [19] was used in the SAYO dataset, and Phred-scaled Quality Score (QUAL) [19] was used in the SPARK dataset for variant quality scores. Our analysis was restricted to rare variants with a maximum allele frequency of 0.0001 in both the gnomAD and UKBB databases.

To correct for potential technical differences and ensure that the two datasets are comparable without systematic inflation or deflation of association signals, we utilized rare synonymous and in-frame insertion/deletion (indel) variant rates as neutral benchmarks. Since these variants are generally functionally neutral and less prone to different subpopulations, their rates should be similar across datasets. We adjusted the variant quality and allele balance scores to minimize the difference in variant rates between the two datasets. We also visually evaluated gene-based variant burden using quantile-quantile (QQ) plots of observed versus expected *p* values with 95% confidence intervals [28,29].

### 2.6. Known Risk Gene-Set Test

We conducted a gene-set burden test to validate our method by evaluating EOCRC associations with known CRC-related genes. Potentially deleterious missense variants were selected using a VEST4 rank score ≥ 0.5, as this threshold reflects a stronger association with cancer [30]. We excluded LoF variants in oncogenes from the analysis, as those variants are protective against cancer [31]. We first analyzed a panel of 21 CRC risk genes (Appendix A) according to the American Society of Clinical Oncology (ASCO) guideline [32]. As only one oncogene exists in this gene set, we expanded our risk gene set to 42 (Appendix A) using the Online Mendelian Inheritance in Man (OMIM) database [33]. The OMIM database contained 21 oncogenes and 21 tumor suppressor genes (TSGs). Notably, 13 genes overlapped between the ASCO and OMIM lists. Statistical significance was evaluated using a binomial test, and the ratio of deleterious variant burdens per individual was used to measure burden enrichment.

### 2.7. Individual Risk Gene Test

We identified individual CRC risk genes using a similar approach to our gene-set analysis. As the distribution of VEST4 predicted scores varies significantly across genes, we employed a variable threshold method [34] to optimize thresholds for classifying deleterious missense variants by minimizing *p* values from binomial tests. The *p* values were then corrected using a permutation test with 10 million iterations, where the case–control status was randomly shuffled. The percentage of iterations with a lower *p* value than the original one from the binomial tests was set as the corrected *p* value. This approach maximizes statistical power and has been shown to be effective in previous genetic studies [35,36]. We conducted burden analyses based on different mechanisms, including oncogenes (gain-of-function (GoF) variants) and TSGs (LoF variants). Given that missense variants can act as either GoF or LoF, we analyzed the variants in different groups: missense only, LoF only, and combined missense and LoF. Deleterious missense variants were defined using different VEST4 thresholds in individual genes. We first tested the entire SAYO case–cohort (comprising CRC and significant polyps) and then further tested a more homogeneous case–cohort with CRC only. In total, we conducted six analyses per gene. To account for multiple testing across different cohorts and variant types, we applied the Bonferroni correction. We also displayed variant distributions along proteins, if protein domain information is available from UniProt [37], using lollipop plots.

### 2.8. Validation in an Independent Whole-Genome Sequencing (WGS) Dataset

To validate our findings, we performed replication analyses on significant genes using the European-ancestry subset of the UKBB [25] whole-genome sequencing (WGS) [38] cohort. CRC phenotypes were defined based on International Classification of Diseases, 10th Revision (ICD-10) codes [39], with all ICD-9 codes converted to their corresponding ICD-10 equivalents. Since majority of SAYO participants have a distal colon, we identified two case–cohorts: proximal colon cancer includes malignant neoplasm of cecum and ascending/transverse colon (ICD-10 codes of C18.0, C18.2, C18.3, C18.4); distal colon cancer includes malignant neoplasm of descending/sigmoid colon (C18.5, C18.6, C18.7), rectosigmoid junction (C19), and rectum (C20). Participants without any of the codes in C18–20 were employed as controls.

We applied a similar variant QC pipeline (Appendix A), restricting to rare variants with a maximum allele frequency of 0.0001 in both gnomAD and UKBB, a genotype quality of greater than 25, an allele depth for alternative alleles of greater than 3, a genotype depth of greater than 10, and a minimum allele balance of 0.2. Variant annotation was performed using the same VEP pipeline. For burden tests, the two CRC sub-cohorts were analyzed separately. We first applied the same VEST4 threshold identified from the SAYO dataset to validate the gene signal and then used the same variable threshold testing to examine the maximal signal from the UKBB dataset.

## 3. Results

### 3.1. Cohort Characteristics

Following the inclusion protocol described in the method, Figure 1A shows the number of individuals excluded under each criterion. Figure 1B shows a PCA plot of the SAYO cohort with predicted ancestry information.

The entire SAYO cohort (n = 270) had a mean age of 41.9 ± 10.7 years, comprising 121 males (44.8%) and 143 females (53.0%), with the remainder not reporting their gender (Table 1).

Based on genetic inference, 255 participants (94.4%) were of European ancestry, aligning closely with the 96.7% who self-identified as Caucasian. The CRC group (n = 193) had a mean age of 42.6 ± 9.3 years, comprising 93 (48.2%) males and 96 (49.7%) females, with 180 (93.3%) self-identifying as Caucasian. The polyp group (n = 44) had a mean age of 31.4 ± 14.1 years, comprising 17 (38.6%) males and 26 (59.1%) females, with 43 (97.7%) identifying as Caucasian. Individuals with non-CRC conditions, such as appendiceal neoplasms, were removed. Excluding related individuals and those of non-European ancestry, the final cohort consisted of 212 cases (174 with CRC and 38 with significant polyps) and 31,699 controls from SPARK. Among all 174 CRC cases, 83 (47.7%) reported family histories of CRC, and 143 (82.2%) reported family histories of any cancer.

### 3.2. Cleaned Datasets

To mitigate batch effects and technical artifacts, we used rates of rare synonymous and in-frame indel variants as metrics to calibrate quality filters. Variants were retained with VQSLOD greater than 2 in SAYO and QUAL greater than 38 [40], and allele balance greater than 0.1 in SAYO and SPARK (Appendix A). The mean number of synonymous variants per individual was 26.43 in the SAYO cohort and 26.65 in the SPARK cohort (Appendix A), resulting in a nonsignificant ratio of 0.99 (*p* value of 0.27). For in-frame indel variants, the variant ratio was 1.07 (*p* value of 0.30). A QQ plot testing rare synonymous and in-frame indel variant rates for individual genes showed no significant deviation between the observed and expected *p* values, suggesting no significant technical bias (Appendix A).

### 3.3. Association of Rare Variants in Known CRC Risk Genes

We identified 36 unique predicted deleterious variants in the ASCO or OMIM-defined CRC risk genes carried by 33 European-ancestry cases (26 with CRC and 7 with significant polyps), shown in Appendix A. Three CRC cases carry two deleterious variants in different genes. Notably, two individuals had only one heterozygous variant in the *MUTYH gene*, which is an autosomal recessive gene. In individuals who had not progressed to CRC, serrated-type polyps were detected in every case. The average age at diagnosis for CRC cases with at least one deleterious variant in a known CRC risk gene is 39.0 years, indicating a trend towards a younger age, but it is not significantly different from the entire European-ancestry CRC population (41.7 years, *p* value = 0.06).

The gene-based burden test revealed that LoF variants in known CRC TSGs were significantly enriched in the SAYO dataset compared to the SPARK dataset. In ASCO-listed genes, the relative risk (RR) for LoF variants was 7.21 (*p* value of 5.8 × 10^−6^), and in OMIM-listed genes, the RR was 3.16 (*p* value of 0.01) (Table 2), reinforcing the strong link between LoF variants and CRC risk across multiple gene sets.

When combining LoF variants with missense variants predicted deleterious by VEST4 (≥0.5), the RR decreased (ASCO: 1.66, *p* value of 0.03; OMIM: 1.20, *p* value of 0.38). Notably, missense variants alone showed no significant enrichment in OMIM genes (RR = 1.00, *p* value = 1.00). This suggests that most known CRC susceptibility genes were driven by large-effect-size LoF variants, and the contribution from the missense variants is minimal. Furthermore, the limited representation of oncogenes in the ASCO dataset (n = 1) likely contributed to the low RR observed for missense variants with VEST4 ≥ 0.5 (*p* value of 0.77), reflecting a potential bias toward TSG content. Collectively, these findings, particularly in detecting LoF variant enrichment among CRC TSGs, underscore the effectiveness of our burden analysis approach.

### 3.4. Novel Individual CRC-Risk Gene Discovery

We conducted a rare variant burden analysis to discover novel susceptibility genes for EOCRC based on different mechanisms, including oncogenes (gain-of-function (GoF) variants) and TSGs (LoF variants). Given that missense variants can act as either GoF or LoF, we analyzed the variants in different groups: missense only, LoF only, and combined missense and LoF. VEST4 thresholds defined deleterious missense variants in individual genes. One set of analyses is based on the entire SAYO dataset (comprising both CRC and polyp cases), and the other set is based on the confirmed CRC cases. Figure 2 presents QQ plots for the burden test for all six analyses. The diagonal line represents the null hypothesis of uniformly distributed *p* values, while genes above the diagonal line indicate an enriched burden of deleterious variants in cases compared with controls.

Using a Bonferroni-corrected significance threshold (*p* value of 2.7 × 10^−6^) and a borderline significant threshold (*p* value of 5.5 × 10^−5^), seven genes showed statistically significant or borderline significant associations in at least one test: *MEIKIN*, *STK25*, *PGBD4*, *DIRAS3*, *ATG3*, *RPS6KA4*, and *DDX42*. *MEIKIN* was consistently associated with CRC across most tests and cohort subsets. Additional genes showed variant-type- or cohort-specific potential associations, as summarized below (Table 3). All individuals who had not progressed to CRC have serrated-type polyps.

*MEIKIN* showed significant enrichment in all variant-based tests, including combined (*p* value of 1.0 × 10^−7^), missense variants-only (*p* value of 4.2 × 10^−5^), and LoF-only (*p* value of 6.3 × 10^−5^) analyses. The Bonferroni method identified *MEIKIN* as the only significant risk gene. Four individuals in the SAYO cohort harbored predicted deleterious variants in *MEIKIN* (2 missense variants, 2 LoFs). Two were CRC cases, and the other two had significant polyps. Three of them—one CRC case and two with significant polyps—had sessile serrated adenomas. The two individuals with significant polyps had a family history of CRC or polyps; one CRC patient reported a family history of Melanoma. The phenotypic heterogeneity may explain why *MEIKIN* has been overlooked in previous genetic studies. One CRC case also carried a frameshift variant in a known CRC risk gene, *POLE* (detailed in Appendix A). *POLE*-associated CRC risk is primarily conferred by pathogenic missense variants; thus, the contribution of this LoF variant remains uncertain.

*STK25* showed an association with LoF variants, suggesting its potential role as a TSG in EOCRC. We identified three LoF variants in CRC cases (Appendix A, Figure 3). Notably, one CRC case also carried a predicted deleterious missense variant in *MLH3*, a gene listed among known CRC-associated genes in the OMIM database but not in the ASCO guidelines. Based on recent ClinGen expert panel reviews, *MLH3* currently has only limited evidence supporting its role in hereditary CRC predisposition. Therefore, while its contribution cannot be ruled out entirely, the *STK25* LoF variants could potentially explain the germline risk factor in this case. Two individuals had first-degree family histories of CRC and/or polyps, and all three had family histories of other cancer types.

*PGBD4* was identified as a potential oncogene based on its association with missense variants only. Four deleterious missense variants and no LoF were identified in the SAYO dataset, while both missense and LoF were found in SPARK. Two individuals with *PGBD4* variants were diagnosed with CRC. Appendix A provides detailed information on individual variants. Three already have one deleterious missense variant in known CRC risk genes—*MCC*, *MSH6*, and *FGFR3*. These three genes are primarily implicated in CRC through somatic variants in tumors or Lynch syndrome, rather than germline missense variants.

Other notable findings include *DIRAS3*. We identified two LoF variants and one deleterious missense variant in *DIRAS3* in the SAYO dataset (detailed in Appendix A), suggesting a TSG, consistent with previous findings [41,42]. All three individuals carrying deleterious *DIRAS3* variants were diagnosed with CRC. The *DIRAS3* signal in controls is driven by a recurrent missense variant, as shown in Figure 3, which typically suggests a benign effect. The association signal could be further improved with more accurate predictors of missense deleteriousness.

*ATG3* had four deleterious missense variants and no LoF variants in the SAYO dataset, while both variant types were identified in SPARK, so it is identified as a potential oncogene. Two individuals with the *ATG3* deleterious variant were diagnosed with CRC; the other two had significant polyps (Appendix A). Three had family histories of CRC and/or significant polyps, and all four had family histories of non-CRC cancer. Four *ATG3* missense variants were identified in controls, primarily localized to two protein regions (Figure 3). This highlights the importance of predicting the functional effects of variants in novel risk gene discovery.

*RPS6KA4* was associated with LoF variants and is a potential TSG. We identified one LoF and two deleterious missense variants in the SAYO dataset. All three individuals are confirmed CRC cases (Appendix A). One CRC case also has a deleterious missense variant in *POLE*, a TSG [43] for CRC [33]. In Figure 3, the *RPS6KA4* variants in cases and controls do not overlap, suggesting that real pathogenic variants may localize in specific protein regions.

*DDX42* showed significant enrichment of risk variants in CRC cases (Appendix A), with a relative risk of 61.1 and a *p* value of 5.4 × 10^−5^ at a VEST4 threshold of 0.88, as shown in Table 3. In Figure 3, all three variants in cases cluster within biologically critical regions: two missense variants fall in the Helicase ATP-binding domain, and one stop-gained variant lies near the Helicase C-terminal domain, where truncation is likely disruptive.

The 17 CRC cases carrying deleterious variants in our seven candidate risk genes had an average age of diagnosis of 41.9 years, which was not significantly different from the overall CRC cohort (*p* value of 0.92), suggesting that these variants may contribute to CRC risk without markedly altering age of onset. However, four individuals with an additional deleterious variant in a known CRC risk gene were diagnosed at a significantly younger average age of 31.5 years (*p* value of 0.04), suggesting a possible additive genetic effect. A family history of CRC or significant polyps was reported in 11 of the 17 CRC cases. Of the 17 CRC cases, 7 had a family history of CRC, and 14 had a family history of any cancer. A statistical comparison using a binomial test revealed no significant differences between the entire CRC cohort (*p* values of 0.6 and 1.0). Notably, although these variants were identified in individuals of European ancestry, no carriers were observed among the non-European cases in SAYO (n = 25).

We further examined tumor MMR immunohistochemistry (IHC) status to explore the underlying pathways in these carriers, which was available for 14 of the 17 cases. Among these, 12 tumors were MMR-proficient, showing intact expression of *MLH1*, *PMS2*, *MSH2*, and *MSH6*. Two cases exhibited MMR deficiency: one showed loss of *MLH1* and was clinically diagnosed with Lynch syndrome (germline *MLH1*: T117M), while the other (a 53-year-old woman) had concurrent loss of *MLH1* and *PMS2* without detectable germline variants in either gene. A somatic pathogenic *BRAF* V600E variant was identified in this woman, suggesting *MLH1* promoter hypermethylation [44], although methylation testing was not available to confirm this. These findings indicate that most variant heterozygotes had MMR-proficient tumors, supporting the involvement of non-MMR pathways in EOCRC susceptibility in this cohort.

### 3.5. Validation of MEIKIN in the UKBB WGS Cohort

We extended our analysis to the UKBB WGS dataset to evaluate *MEIKIN* effects in an independent population-scale cohort. The UKBB includes 502,242 participants, of whom 472,462 remained after restricting the analysis to individuals of European ancestry. Cancer phenotypes were defined using ICD-10 codes, with all ICD-9 codes converted to their corresponding ICD-10 equivalents: proximal colon cancer (n = 2816), distal colon cancer (n = 2557), rectosigmoid junction cancer (n = 619), and rectal cancer (n = 2596). The average age at diagnosis was 66.51, 63.98, 62.12, and 63.38 years for proximal colon, distal colon, rectosigmoid junction, and rectal cancer, respectively. Under-55 cases comprise 9.7%, 15.9%, 21.9%, and 18.7% of each group (Appendix A).

Following application of the same rare variant filters as in the SAYO cohorts, a total of 127 unique rare coding variants were retained in *MEIKIN*. *MEIKIN* LoF variants were enriched in distal colon cases, demonstrating a relative risk of 2.46 and a *p* value of 0.086 in Appendix A. Applying the deleteriousness threshold identified in the SAYO dataset (VEST4 ≥ 0.109) and the variable-threshold approach (VEST4 ≥ 0.24), missense variants were insignificantly enriched in distal colon cases. When combining missense and LoF variants, the variable-threshold approach identified a borderline significance with the relative risk of 1.98 and a *p* value of 0.048, as shown in Appendix A. Analyzing distal colon cases in individuals younger than 55 years, the variable-threshold approach revealed an insignificant enrichment, with a relative risk of 8.23 and a *p* value of 0.217. No significant enrichment was observed in proximal colon cancer. Although full replication of the *MEIKIN* association across colorectal subtypes was not achieved, these results suggest that *MEIKIN* missense and LoF variants are potentially associated with distal colon cancer.

## 4. Discussion

Our gene-based rare-variant analysis identified one novel candidate risk gene, *MEIKIN*. It is a kinetochore protein and plays a crucial role in meiosis [45,46,47]. In addition to the testis, *MEIKIN* is not highly expressed in any other somatic tissues. However, ectopic activation of meiotic genes has been increasingly recognized as a contributor to oncogenesis, disrupting normal cellular processes, chromosome cohesion, kinetochore function, and segregation fidelity, contributing to cancer hallmarks [48,49,50,51,52]. Based on STRING analysis [53], *MEIKIN* is implicated in chromosome segregation and cell cycle regulation [45], through interactions with key proteins such as SGO1 [54], SGO2 [54], PLK1 [55], and ESPL1 [56], all of which are known to play critical roles in cancer-associated pathways (Appendix A). Moreover, we demonstrated the enriched *MEIKIN* missense and LoF variants in the distal colon cancer cases in the European-ancestry subset of the UKBB WGS cohort. This replication in an independent, population-scale dataset reinforces the robustness of our analytic framework and provides additional evidence that rare LoF variation in meiotic regulators may contribute to EOCRC susceptibility. Notably, our SAYO datasets defined case status based on clinical diagnoses of CRC, whereas the UKBB relies on ICD billing codes. The UKBB primarily consists of older adults; analyses focused on early-onset diseases were limited. Differences in disease ascertainment and age composition in cases between the two cohorts may partially explain the lack of statistical significance. Nonetheless, our replication in the distal colon cancer cases aligns with the fact that EOCRC is more likely to occur in the distal colon.

Our study extends previous work by identifying additional candidate genes with diverse biological functions that may contribute to EOCRC susceptibility [57,58], including *STK25*, *PGBD4*, *DIRAS3*, *ATG3*, *RPS6KA4*, and *DDX42*. *STK25* is critical in CRC progression by regulating autophagy, metabolism, and epithelial–mesenchymal transition (EMT) [59,60,61]. Its downregulation enhances autophagy via the *JAK2/STAT3* pathway, while higher *STK25* levels are associated with a better prognosis [59]. *STK25* also inhibits CRC cell proliferation and glycolysis through interaction with *GOLPH3* and modulation of the mTOR pathway [60]. Additionally, *STK25* promotes EMT by interacting with *LIMK1*, contributing to increased invasion and poorer outcomes [61]. The enrichment of LoF variants in *STK25* among cases supports its putative role as a CRC tumor suppressor. While little is known about *PGBD4*, other members of the PGBD gene family have been more extensively characterized. Notably, *PGBD5* has been shown to drive site-specific oncogenic mutations in human tumors, implicating it in tumorigenesis [62]. Although the biological role of *PGBD4* itself is undercharacterized, its enrichment signal in EOCRC and its close homology with the oncogenic *PGBD5* gene suggest potential functional relevance. This highlights the diverse and impactful roles that PGBD family genes may play in the development and progression of cancer. *DIRAS3* plays a crucial role in inhibiting RAS/MAPK signaling, which is often dysregulated in many cancers. *ATG3* was upregulated in CRC tissues and cell lines compared to normal counterparts, with experimental evidence showing that *ATG3* knockdown significantly suppresses cancer cell proliferation and invasion [63]. The oncogenic effects of *ATG3* are primarily mediated through an autophagy-dependent pathway, evidenced by the counteractive effects of autophagy blockade on cancer progression [63]. Additionally, the interaction of *ATG3* with *ATG12* suggests a broader role in the autophagy machinery [64,65]. We conducted AlphaFold 3D structural modeling of the ATG3 missense variants. Two variants in solvent-exposed regions may affect protein interactions, and the other two in buried positions may affect protein folding or local stability. *RPS6KA4* is involved in tumor suppression through epigenetic regulation and interaction with key signaling pathways [66]. Furthermore, high-throughput LoF screening has identified *RPS6KA4* as a possible regulator of p53 activity, reinforcing its involvement in key cancer-associated signaling pathways [67]. *DDX42* encodes a DEAD-box RNA helicase with RNA-chaperone activity that participates in pre-mRNA splicing, RNA remodeling, and maintenance of genome stability [68]. It interacts with *ASPP2*, a co-activator of p53, thereby modulating apoptosis, and has been identified as a PARP1 interactor, further linking *DDX42* to DNA damage response [69]. *DDX42* also functions as an intrinsic inhibitor of retroviruses and LINE-1 retrotransposons, restricting aberrant nucleic acid species that can promote genomic instability [70]. In contrast, overexpression of *DDX42* in hepatocellular carcinoma activates the PI3K/AKT signaling pathway, enhancing proliferation, radioresistance, and sorafenib resistance, suggesting context-dependent oncogenic activity [71].

Our study has several strengths. Our analytic approach was validated by the detection of established associations with LoF variants in known TSGs, confirming its robustness. The identification of *ATG3*, *STK25*, and *DIRAS3* aligns with prior research implicating autophagy and tumor suppression mechanisms in CRC development. The use of stringent quality control metrics, a tailored burden test, and 10 million permutations provided high confidence in the robustness of our association findings. Prioritizing predicted deleterious variants further enriched our results for potentially functional contributors to disease risk. The identification of additional candidate genes with diverse biological functions expands the current spectrum of CRC predisposition loci, opening new avenues for biological and clinical investigation.

Nevertheless, certain limitations should be acknowledged. First, our results provide statistical evidence rather than definitive evidence of causality. Functional studies are crucial for validating the impact of these variants. Future studies should focus on experimentally characterizing the candidate genes through gene editing, cellular modeling, and transcriptomic profiling to elucidate their roles in the initiation and progression of CRC. Second, while our sample size was sufficient to detect strong associations, larger and more diverse cohorts will be needed to identify lower-penetrance risk genes and enhance generalizability across different ancestries. Finally, small-effect contributors, including common variants [72,73,74], lifestyle factors, and environmental exposures, were not assessed within our rare-variant association tests.

Clinically, the discovery of novel genetic risk factors could enhance early detection and personalized screening for individuals and families at elevated risk of EOCRC. If validated, these genes may be incorporated into multi-gene panels, improving risk stratification beyond known CRC susceptibility genes. Moreover, insights into the molecular pathways affected by these genes, particularly those involving autophagy, metabolism, and immune signaling, could inform the development of new therapeutic targets. As EOCRC incidence continues to rise globally, understanding the contribution of rare germline variation will be essential to refining both preventive and treatment strategies.

In summary, our findings broaden the landscape of EOCRC-associated genes and highlight the potential of rare variant burden analysis to uncover biologically and clinically relevant risk loci. Continued functional validation and integration with clinical phenotypes will be key to translating these discoveries into precision oncology for EOCRC.

## 5. Conclusions

Using exome sequencing and rare variant burden testing in a well-curated EOCRC cohort, we confirmed significant enrichment of LoF variants in established tumor suppressor genes and identified *MEIKIN* as a novel candidate susceptibility gene, with supportive evidence in an independent population dataset for distal colon cancer. We also identify *STK25*, *PGBD4*, *DIRAS3*, *ATG3*, *RPS6KA4*, and *DDX42* as additional candidates with variant-class-specific signals. Collectively, these findings underscore a meaningful contribution of rare germline variants—both LoF and GoF—to the genetic architecture of EOCRC.

These results expand the landscape of heritable CRC risk and suggest practical avenues for improving genetic risk assessment. Pending replication and functional validation, the genes highlighted here could inform panel design, enable earlier identification of high-risk individuals, and guide mechanistic studies of pathways such as kinetochore biology, autophagy, and immune signaling. Larger, ancestrally diverse cohorts and experimental follow-up will be essential to translate these discoveries into precision screening and prevention strategies.

## Figures and Tables

**Figure 1 cancers-17-03931-f001:**
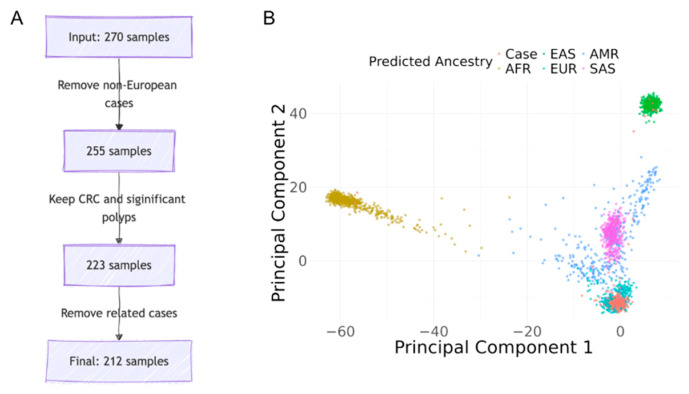
Case selection and ancestry predictions. (**A**) Case selection process: from 270 individuals to the final cohort. (**B**) Principal component analysis for ancestry predictions in the SAYO cohort.

**Figure 2 cancers-17-03931-f002:**
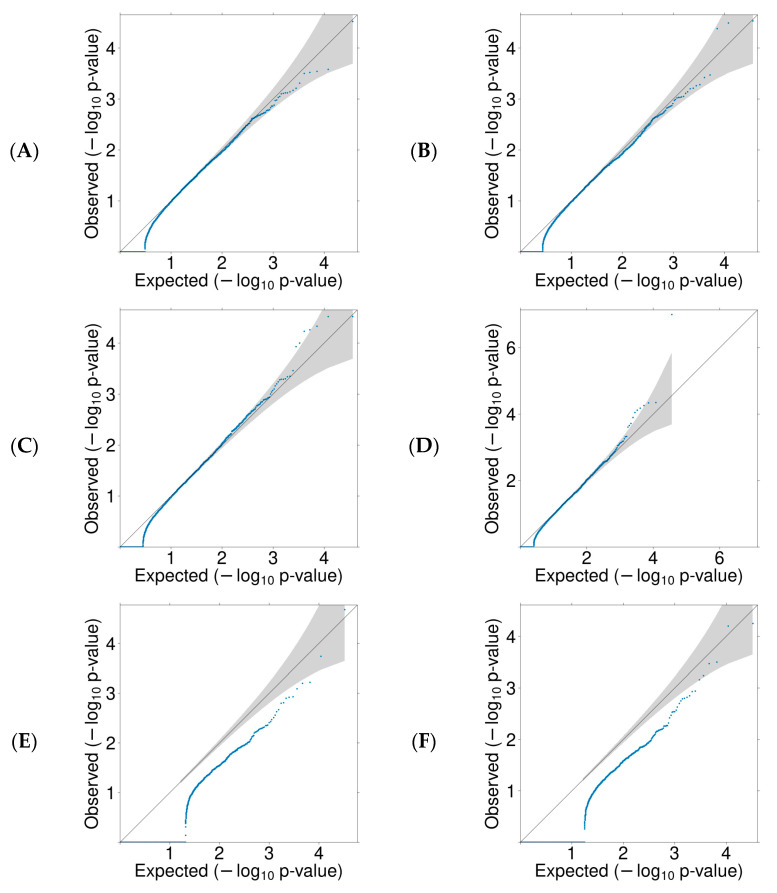
QQ plots for missense and/or LoF variants burden test. Each blue dot represents a gene, and the 95% confidence intervals are shaded. (**A**) missense in CRC, (**B**) missense in CRC and significant polyps, (**C**) missense and LoF in CRC, (**D**) missense and LoF in CRC and significant polyps, (**E**) LoF in CRC, (**F**) LoF in CRC and significant polyps.

**Figure 3 cancers-17-03931-f003:**
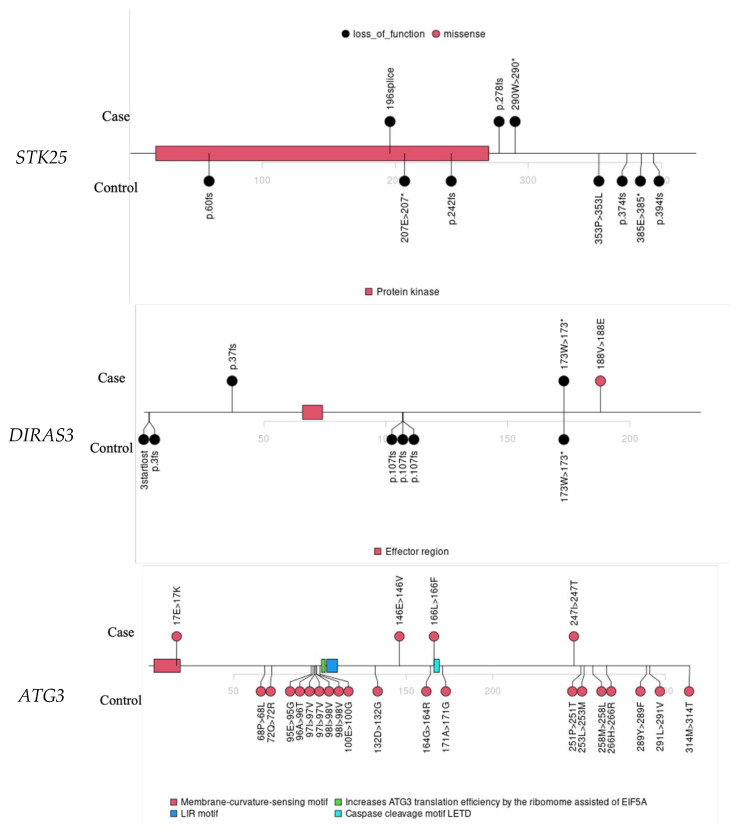
Lollipop plots for variants in *STK25*, *DIRAS3*, *ATG3*, *RPS6KA4*, and *DDX42*.

**Table 1 cancers-17-03931-t001:** Demographics of the SAYO cohort.

Phenotype	Age (Years) ^a^	Gender (%) ^b^	Self Reported Race (%) ^c^
**CRC (193)**	42.6 ± 9.3	M: 93 (48.2) F: 96 (49.7)	Caucasian: 180 (93.3)
**Polys (44)**	31.4 ± 14.1	M: 17 (38.6) F: 26 (59.1)	Caucasian: 43 (97.7)
**All (270)**	41.9 ± 10.7	M: 121 (44.8) F: 143 (53.0)	Caucasian: 255 (94.4)

^a^ Mean value ± Standard Deviation. ^b^ Number (Percentage) of Male (M) and Female (F). ^c^ Number (Percentage) of Caucasians.

**Table 2 cancers-17-03931-t002:** Deleterious variant burden in 21 and 42 CRC-associated genes from ASCO and OMIM.

Gene Set	Variant Type	# of Variants per Individual in SAYO	# of Variants per Individual in SPARK	Relative Risk	*p* Value
**ASCO**	LoF	0.04	0.01	7.21	5.8 × 10^−6^
	Missense with VEST4 ≥ 0.5	0.06	0.05	1.05	0.77
	Missense with VEST4 ≥ 0.5 + LoF	0.10	0.06	1.66	0.03
**OMIM**	LoF	0.03	0.01	3.16	0.01
	Missense with VEST4 ≥ 0.5	0.09	0.09	1.00	1.00
	Missense with VEST4 ≥ 0.5 + LoF	0.11	0.10	1.20	0.38

# means number.

**Table 3 cancers-17-03931-t003:** Top EOCRC risk genes identified in case–control burden analysis.

Gene	# of Variants per Individual in Cases	# of Variants per Individual in Controls	Relative Risk	VEST4 Threshold	*p* Value	Most Significant Analysis
** *MEIKIN* **	0.019	0	150.5	0.11	1.0 × 10^−7^	LoF and missense in CRC and polyps
** *STK25* **	0.017	2.2 × 10^−4^	55.0	1.00	2.1 × 10^−5^	LoF in CRC
** *PGBD4* **	0.019	5.4 × 10^−4^	28.7	0.23	2.9 × 10^−5^	missense in CRC and polyps
** *DIRAS3* **	0.017	1.9 × 10^−4^	61.1	0.94	3.0 × 10^−5^	LoF and missense in CRC
** *ATG3* **	0.019	6.0 × 10^−4^	26.2	0.51	3.2 × 10^−5^	missense in CRC and polyps
** *RPS6KA4* **	0.017	1.9 × 10^−4^	61.1	0.86	4.7 × 10^−5^	LoF and missense in CRC
** *DDX42* **	0.017	1.9 × 10^−4^	61.1	0.88	5.4 × 10^−5^	LoF and missense in CRC

# means number.

## Data Availability

Exome sequencing data from the SAYO study will be made available upon request.

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
