# Peer review of "Identification of Novel Susceptibility Genes for Early-Onset Colorectal Cancer Through Germline Rare Variant Burden Testing"

_cancers, 2025, doi:10.3390/cancers17243931_

Round 1

Reviewer 1 Report

Comments and Suggestions for Authors

Major Revision Recommendation:

The manuscript presents valuable insights into EOCRC genetics; however, substantial revisions are required to strengthen methodological rigor and interpretability. The authors should provide deeper justification for variant-quality calibration, expand functional context for newly implicated genes (e.g., MEIKIN, STK25), and clarify cohort selection criteria with enhanced statistical transparency. Additional validation analyses and more explicit discussion of cross-ancestry generalizability are needed. Improving figure interpretability and aligning conclusions more tightly with presented evidence will substantially enhance the manuscript’s impact.

  1. provide more detail on clinical phenotyping, especially regarding serrated vs adenomatous pathways among variant carriers?
  2. Were lifestyle or environmental covariates (diabetes, obesity, diet) considered in variant burden analyses, given their known association with EOCRC?
  3. The calibration relies on synonymous and inframe indels. Can the authors show these metrics across additional variant categories (e.g., UTR, intronic splice region)?
  4. The use of a variable VEST4 threshold is appropriate, but can the authors report how often the threshold was near zero (risk of overfitting)?
  5. MEIKIN is a meiotic gene; can the authors provide transcriptomic evidence (GTEx, TCGA) of its expression in colon tissues?
  6. For STK25, can the authors experimentally or computationally assess whether the C-terminal LoF variants in controls escape NMD?
  7. ATG3 was labeled as a potential oncogene based on missense enrichment. Can the authors analyze these variants using protein structural modeling?
  8. DIRAS3 variants appear both LoF and missense in cases; can the authors clarify whether tumor sequencing reveals biallelic inactivation?
  9. CD40 variants were mostly missense. Can the authors evaluate impacts on immune signaling pathways or antigen presentation?

Author Response

Major Revision Recommendation:

The manuscript presents valuable insights into EOCRC genetics; however, substantial revisions are required to strengthen methodological rigor and interpretability. The authors should provide deeper justification for variant-quality calibration, expand functional context for newly implicated genes (e.g., MEIKIN, STK25), and clarify cohort selection criteria with enhanced statistical transparency. Additional validation analyses and more explicit discussion of cross-ancestry generalizability are needed. Improving figure interpretability and aligning conclusions more tightly with presented evidence will substantially enhance the manuscript’s impact.

  1. Provide more detail on clinical phenotyping, especially regarding serrated vs adenomatous pathways among variant carriers?

Thank you for this helpful suggestion. We have added detailed clinical phenotyping information, including tumor location and type of polyps, for individuals carrying predicted pathogenic variants in known CRC risk genes (Table S4) and in proposed risk genes (Table 5). Among individuals with polys who have not developed CRC, all presented with serrated-type polyps. This information has been incorporated into the Result.

  1. Were lifestyle or environmental covariates (diabetes, obesity, diet) considered in variant burden analyses, given their known association with EOCRC?

We acknowledge the importance of lifestyle or environmental covariates in the genome-wide association studies, where the genotype effects are relatively small. Our analysis focuses on rare variants with big effect sizes; lifestyle or environmental factors (such as diabetes, obesity, or diet) have relatively small effects compared to the pathogenic variants. In this setting, lifestyle exposures typically have substantially smaller effect sizes, and adjusting for them has a limited influence on gene-based rare variant signal detection. Our case and control cohorts were assembled from multiple sources with incomplete and non-uniform lifestyle and metabolic profiles, making harmonization infeasible without introducing systematic bias. Thus, these variables were not used as statistical covariates in the rare variant burden testing. Nevertheless, we acknowledge that lifestyle factors contribute to overall EOCRC risk and have added this limitation in the Discussion.

  1. The calibration relies on synonymous and inframe indels. Can the authors show these metrics across additional variant categories (e.g., UTR, intronic splice region)?

Thank you for the suggestion. Our calibration strategy focuses on synonymous variants and inframe indels because these categories are well-established neutral baselines for quality assessment in whole-exome sequencing data, where coverage, capture efficiency, and variant calling accuracy are highest, making them suitable for assessing technical calibration and call-level reliability. In contrast, variants within UTRs and intron regions are largely outside the capture design and therefore not reliably detected in our dataset. For this reason, we did not include those categories in the calibration analysis.

  1. The use of a variable VEST4 threshold is appropriate, but can the authors report how often the threshold was near zero (risk of overfitting)?

We thank the reviewer for raising this important point. We provided the distribution of VEST4 thresholds across 18,247 genes included in our rare-variant burden analysis. As shown in the histogram below, most genes had thresholds close to 1.0. These results indicate that the variable threshold procedure was stable and did not rely on arbitrarily small thresholds.

  1. MEIKIN is a meiotic gene; can the authors provide transcriptomic evidence (GTEx, TCGA) of its expression in colon tissues?

We thank the reviewer for this important point. Consistent with prior reports, MEIKIN exhibits low expression across most normal somatic tissues, including the colon, as shown in GTEx, HubMap, and related transcriptomic resources. We have added this clarification to the revised Discussion to ensure that the baseline expression pattern is clearly described. However, multiple studies have demonstrated that meiosis-specific or germline-restricted genes can become aberrantly re-expressed in somatic cancers, contributing to chromosomal instability and tumor progression. This phenomenon, commonly referred to as cancer-testis gene activation, has been well documented (Sou et al., 2023; Lingg et al., 2022; Bruggeman et al., 2023). We have added clarifying text in the Discussion.

  1. For STK25, can the authors experimentally or computationally assess whether the C-terminal LoF variants in controls escape NMD?

We appreciate the reviewer’s comment and have reviewed the LOFTEE for those LoF variants. All of them are high-confidence variants, so we have removed our statement about the nonsense-mediated decay.

  1. ATG3 was labeled as a potential oncogene based on missense enrichment. Can the authors analyze these variants using protein structural modeling?

We thank the reviewer for this constructive suggestion. We annotated four ATG3 missense variants using the AlphaFold-predicted ATG3 (Q9NT62) protein model, extracting both the corresponding secondary structure and solvent accessibility at the corresponding locations. The results are summarized below. These structural annotations suggest that the ATG3 missense variants identified in cases occur in distinct structural environments, including residues located in α-helices, coils, turns, and β-strands, with variable solvent exposure. Notably, the two variants at solvent-exposed regions (positions 146 and 166) may be more likely to affect protein-protein interactions or regulatory interfaces, whereas variants at more buried positions (positions 17 and 247) may have a greater potential to perturb folding or local stability. We have added this description to the manuscript.

Variant

Amino acid position

Secondary structure

Solvent accessibility

chr3:112561480 C>T

17

H (helix)

0.1173

chr3:112541841 T>A

146

C (coil)

0.5132

chr3:112538158 C>G

166

T (turn)

0.3226

chr3:112536529 A>G

247

E (beta-strand)

0.1026

  1. DIRAS3 variants appear both LoF and missense in cases; can the authors clarify whether tumor sequencing reveals biallelic inactivation?

We appreciate the reviewer’s question. In our dataset, each DIRAS3 carrier harbored only a single germline variant, and matched tumor sequencing data were not available. Therefore, we are unable to assess potential biallelic inactivation in tumor tissue.

  1. CD40 variants were mostly missense. Can the authors evaluate impacts on immune signaling pathways or antigen presentation?

We appreciate the reviewer’s suggestion. While our analysis prioritized predicted deleterious variants, current computational tools do not reliably determine pathway-specific functional impacts, such as effects on immune signaling or antigen presentation. As our study was designed to evaluate genetic association rather than mechanistic evaluation, we were unable to experimentally assess downstream functional consequences. Nonetheless, we agree that elucidating the role of CD40 missense variants in immune pathways represents an important avenue for future investigation, and we have highlighted this in the Discussion as a direction for future functional investigation.

Reviewer 2 Report

Comments and Suggestions for Authors

The authors are commended for their meticulous, batch-effect-calibrated rare variant burden analysis, which robustly confirmed known genetic associations and led to the highly significant discovery of MEIKIN as a novel early-onset colorectal cancer susceptibility gene. The manuscript is well written in terms of providing the latest references also. However, I have following critical comments which I want the authors to address carefully before the acceptance of the manuscript.  

  1. Functional Causality: The strong statistical association identified for MEIKIN requires immediate functional validation to definitively establish biological causality and move beyond a statistical correlation.

  2. Replication and Generalizability: Urgent replication studies are needed in ancestrally diverse cohorts to confirm the generalizability of these European-ancestry-specific EOCRC susceptibility findings.

  3. Secondary Candidate Refinement: The study must include critical functional assessment for genes like STK25 to differentiate truly pathogenic LoF variants in cases from potentially benign C-terminal variants observed in controls.

  4. Mechanistic Validation: The proposed mechanism linking MEIKIN germline variants to kinetochore assembly and chromosomal instability needs experimental verification, such as quantifying aneuploidy rates in cell models.

  5. Clinical Model Integration: The observation that carriers of multiple rare variants have a significantly younger age of diagnosis strongly supports an oligogenic risk model that must be incorporated into future clinical risk assessment panels.

Comments on the Quality of English Language

Can be further improved. 

Author Response

The authors are commended for their meticulous, batch-effect-calibrated rare variant burden analysis, which robustly confirmed known genetic associations and led to the highly significant discovery of MEIKIN as a novel early-onset colorectal cancer susceptibility gene. The manuscript is well written in terms of providing the latest references also. However, I have following critical comments which I want the authors to address carefully before the acceptance of the manuscript. 

  1. Functional Causality: The strong statistical association identified for MEIKIN requires immediate functional validation to definitively establish biological causality and move beyond a statistical correlation.

We agree that functional studies are ultimately required to establish biological causality. However, the scope of this work is a population-level germline rare variant association study, which focuses on statistical correlations rather than experimental validation. We have now clarified in the Discussion that, while our findings provide statistical evidence and mechanistic plausibility for some candidate genes, definitive functional validation will require targeted experimental studies that fall beyond the scope of the current analysis.

  1. Replication and Generalizability: Urgent replication studies are needed in ancestrally diverse cohorts to confirm the generalizability of these European-ancestry-specific EOCRC susceptibility findings.

We agree that replication in ancestrally diverse cohorts is essential for assessing the generalizability of these findings. However, UK Biobank, and therefore our analytic sample, is predominantly of European ancestry. Currently, sufficiently powered rare-variant EOCRC datasets are limited in non-European populations, which prevents us from performing multi-ancestry replication within this study. There is a need for future studies incorporating multi-ancestry cohorts to validate the observed associations. We have included this point in the Discussion.

  1. Secondary Candidate Refinement: The study must include critical functional assessment for genes like STK25 to differentiate truly pathogenic LoF variants in cases from potentially benign C-terminal variants observed in controls.

We appreciate the reviewer’s suggestion and have reviewed the LOFTEE for those LoF variants. All of them are high-confidence variants. For STK25 LoF variants in controls, we have removed our statement about the nonsense-mediated decay.

  1. Mechanistic Validation: The proposed mechanism linking MEIKIN germline variants to kinetochore assembly and chromosomal instability needs experimental verification, such as quantifying aneuploidy rates in cell models.

We agree that experimental verification of the proposed mechanism would be valuable. However, such mechanistic assays are beyond the scope of this study, which focuses solely on population-level associations of germline variants. Future experimental work will be required to validate the mechanistic link between MEIKIN variants and chromosomal instability.

  1. Clinical Model Integration: The observation that carriers of multiple rare variants have a significantly younger age of diagnosis strongly supports an oligogenic risk model that must be incorporated into future clinical risk assessment panels.

We appreciate and agree with the reviewer’s insightful comment regarding potential oligogenic effects. Our finding is that carriers of multiple rare variants present at a younger age at diagnosis, but this difference is insignificant compared to the entire CRC cases within the SAYO cohort. This observation is based on a single cohort. Larger genomic studies and independent validation will be needed to further investigate the influence of the oligogenic risk model on cancer onset.

Round 2

Reviewer 1 Report

Comments and Suggestions for Authors

Accept in present form

Reviewer 2 Report

Comments and Suggestions for Authors

Manuscript can now be accepted in current form.

Comments on the Quality of English Language

Can be further improved.